# Moral Identity and Subjective Well-Being: The Mediating Role of Identity Commitment Quality

**DOI:** 10.3390/ijerph18189795

**Published:** 2021-09-17

**Authors:** Peng Cui, Yanhui Mao, Yufan Shen, Jianhong Ma

**Affiliations:** 1Department of Psychology and Behavior Sciences, Zhejiang University, Hangzhou 310058, China; cuip@zju.edu.cn; 2School of Educational Science, Zhoukou Normal University, Zhoukou 466001, China; Shenyufan2006@126.com; 3Institute of Applied Psychology, Psychological Research and Counseling Center, Southwest Jiaotong University, Chengdu 610031, China

**Keywords:** moral identity, subjective well-being, identity commitment quality, internalization, symbolization

## Abstract

Moral identity is associated with people’s subjective well-being; however, little is known about how an individual with moral identity relates to one’s subjective well-being. Based on the eudaimonic identity theory, the current study proposed that identity commitment quality is a critical mechanism that links moral identity (two dimensions: internalization and symbolization) and subjective well-being. We examined our hypotheses in 419 college students, who completed the Self-importance of Moral Identity Questionnaire, Satisfaction with Life Scale, Scale of Positive and Negative Experience, and Questionnaire for Eudaimonic Well-being. Results confirmed significant positive correlations among moral identity, identity commitment quality, and subjective well-being; findings also suggested that both the internalization and symbolization dimensions of moral identity predicted subjective well-being through identity commitment quality, and identity commitment quality fully mediated the pathway relationship between moral identity and subjective well-being. We discussed these findings with respect to implications and proposed research suggestions for future studies.

## 1. Introduction

In the past decade, empirical research on happiness and well-being has grown enormously [1]. Morality is believed to be strongly associated with happiness [2]; even young children believe that a moral person would be happier [3]. Empirical evidence on the link between morality and happiness has been well-documented in research records [4,5], of which some have suggested that engaging in moral behavior can bring us happiness (for a review, see [6]). At present, the well-being of adolescents is a topic of concern in various fields [7,8]. The development of adolescents’ identity, especially the integration of moral identity and self-identity, is a direction worthy of researchers’ attention. However, the relation between morality and happiness or well-being remains vaguely understood, especially during the critical period of identity formation in late adolescence and early adulthood. 

In personal and moral development, an important task is to form an integrated moral identity, because morality and identity are two facets of the same development system that are unified in late adolescence [9]. Therefore, identity formation may play a critical role in the relation between moral identity and well-being in this period. However, the exact mechanism by which an important identity-moral identity-is linked to human well-being remains unclear. Therefore, the present study aimed to fill this gap and focus on the relation between moral identity and subjective well-being (SWB) by exploring a possible mechanism between people’s moral identity and their SWB, taking into consideration identity commitment quality as a mediating variable. 

### 1.1. Moral Identity and Well-Being

Moral development is closely related to identity formation, and moral self-identity is crucial for living a purposeful life, thus contributing to one’s well-being [10]. Generally, moral identity is thought to signify the importance or salience of morality in one’s identity [11]. The more a person’s moral identity is central to their sense of true self, the bigger the role it plays in behavior and commitment. From a social cognitive perspective, moral identity is defined as the extent to which being a person with moral traits is a social identity that is salient to one’s self-concept [12,13]. The moral identity model has two sub-dimensions: internalization and symbolization. The former refers to the extent to which moral self-schema is experienced as being central to one’s self-definition, whereas the latter refers to the extent to which the moral person’s social identity is expressed through one’s real-world behavior, such as engaging in self-salient activities [12].

Research in the identity development domain has confirmed the positive association between identity achievement and different kinds of happiness measures [14]. Individuals with a more mature sense of identity have better mental health and psychological well-being [15]. However, identity development research has tended to measure the sense of identity as a unitary construct, paying little attention to the relative importance of identity content on which one’s identity is based. A recent study found that identity commitment and in-depth exploration significantly predicts well-being in many identity domains [16]; however, this research failed to include moral identity. 

Given that human morality is much more essential to identity than personality traits, memory, or desires [17], more attention should be given to the relationship between moral identity and well-being. A few recent studies have investigated the important role of moral identity in predicting human well-being [4,18,19]. For instance, Garcia et al. [4] found that moral identity is associated with meaning, engagement and identification with others, acceptance of others, and the sense of being part of something bigger than the self. Hardy et al. [18] examined the role of moral identity in predicting mental health and meaning, demonstrating that identity maturity interacting with moral identity predicts mental health and life meaning. Moreover, moral identity can predict eudaimonia in youth footballers, especially when their social affiliations are low [19]. Taken together, these studies have either focused only on the internalization aspects of moral identity [18,19] or failed to distinguish internalization from symbolization [19], neglecting the fact that the symbolization dimension of moral identity may also play an important role in promoting well-being.

Meanwhile, studies have found that moral identity can guide people to experience SWB through prosocial behaviors, such as volunteering and charity donation. In other words, engaging in prosocial behaviors has been regarded as an effective way to promote human well-being [20]. Moral identity is thought to be an important psychological antecedent and an individual difference variable related to prosocial behavior. Individuals with a strong moral identity are more likely to volunteer in the community, donate food to help the needy [12], give money to a charity that benefits an out-group [21], and make prosocial business decisions [13]. A recent meta-analysis has also indicated that moral identity is significantly associated with prosocial behaviors [22]. Individuals with a strong moral identity consider being helpful to others important, behave in a prosocial manner out of autonomy motivation, and experience positive affect association with such action [23]. A recent representative cross-sectional study in 166 countries has proven that prosociality is significantly associated with positive affect, one of the important components of SWB [24]. Thus, ‘‘doing good’’ may be an important avenue by which people create meaningful and satisfying lives, and moral identity can be considered a promising variable in predicting SWB. 

Moral identity not only leads to SWB through prosocial behavior, but is also positively associated with affective experiences, such as more positive and less negative affect (for a recent meta-analysis, see [25]). People with higher levels of moral identity internalization have higher levels of sympathy and moral reasoning [12], and have heightened moral elevation, such as positive emotions, positive views of humanity, and the desire to be a better person [26]. Thus, moral identity, when being considered as a personality trait, is associated with SWB. However, the underlying mechanism on how moral identity is related to SWB needs further exploration, given that moral identity promoting SWB has only been considered under specific contexts and specific activity experiences in the life span of typical individuals. 

Considering this research gap, we tested the associations between moral identity and subjective well-being drawing on eudaimonic identity theory [27,28], which states that people who have established identity-defining commitment (versus those who do not have identity commitment) would have greater eudaimonic functioning, especially in discovering and recognizing their daimon or true self through personal salient identity-related activities. Thus, as one of the most important personal identities, the commitment of moral identity can bring eudaimonic function, which would be bound to produce SWB. Therefore, we assumed that the self-importance of moral identity in late adolescence or early adulthood would promote their SWB. Our work intended to expand existing research on the relation between identity formation and psychosocial benefit, which has mainly considered the effect of specific identities [29,30,31]. However, whether such relations can be applied to moral identity remains unclear. Therefore, the present work trying to examine such a relationship between moral identity and SWB is a novel effort in this area.

### 1.2. The Mediating Role of Identity Commitment Quality

Extensive research in the field of identity has provided substantial evidence of a positive relation between identity formation and well-being. This trend of work inherits Ericson’s classical theory of identity development, which highlights exploration and commitment as central processes by which individuals form personal identities [32]. Traditional identity research has fixated on the relationship between identity commitment, exploration, and psychosocial functioning, and a consistent finding of these studies is that status, characterized by identity commitment, is generally associated with higher levels of psychosocial functioning [33]. Identity commitment is also significantly correlated with different kinds of well-being [14]. Individuals who have established identity commitments are more likely to report feelings of personal expressiveness, which is a core concept of eudaimonia [34]. In other words, involvement in identity-related and personal expressive activities is associated with more positive psychosocial outcomes. However, according to eudaimonic identity theory, the establishment of identity commitment does not necessarily bring SWB, and the commitment must be consistent with one’s daimon or true self to have a eudaimonic function [27,28]. In late adolescence and early adulthood, the development of moral identity and the integration of moral identity and self-identity can greatly promote the improvement of eudaimonic functioning [10].

Eudaimonic functioning is also referred to as identity commitment quality in prior research [35,36]. In addition to identity formation and identity content, differences in the quality of identity selection related to adjustment and psychosocial functioning may play a major role in SWB. Soenens et al. [35] examined the role of commitment quality in terms of autonomous and controlled motivation associated with identity commitment based on self-determination theory. A main finding is that, even after controlling for identity commitment, autonomy motivation is still positively associated with adjustment, whereas controlled motivation is negatively related to adjustment, and the quality of identity motivation partially mediates the link between identity styles and adjustment. Drawing upon eudaimonic identity theory, Waterman et al. [36] investigated the quality of identity commitment and found that commitment quality accounts almost entirely for the association of identity commitments with psychosocial functioning. In other words, for people who have low levels of identity commitment, identity commitments are associated with costs instead of benefits. The results of these studies suggest that the quality of commitment has an independent effect on psychosocial benefits other than identity commitment. However, identity formation does not end in adolescence, but continues into adulthood [37]. In late adolescence or early adulthood, moral identity development and self-development undergo integration, and the formation of identity commitment, as well as the quality of identity commitment, may enhance SWB by promoting eudaimonic functioning. However, none of the existing studies have addressed the important relationship between identity commitment quality, moral identity, and SWB. 

Based on eudaimonic identity theory, living in truth to one’s true self gives rise to eudaimonia, which certainly leads to SWB [27,28]. The formation of identity, especially important personal moral identity, will lead first to improvements in eudaimonic functioning, and then to improvement in positive subjective experience. Notably, eudaimonia is not an innate structure but one that requires active development. College students face the process of identity development and integration as they will gradually form their moral identity. One’s defined moral identity captures the importance of morality to one’s self-identity and is an identification with a specific commitment, whereas high-quality commitment reflects optimal psychological functioning. Therefore, in addition to moral identity commitment, the quality of identity commitment plays a more critical role in the subjective experience and cognitive evaluation of happiness. When moral identity commitment is consistent with personal expressiveness, it can promote well-being more than other commitments [36]. 

Therefore, the present study sought to investigate not only the relation between moral identity and SWB, but also the mediating role of identity commitment quality in order to examine the underlying mechanism between moral identity and SWB. We assumed that both the self-importance of moral identity and identity commitment quality contribute to SWB, whereas the self-importance of moral identity would predict SWB through the mediation effect of identity commitment quality.

### 1.3. The Role of Internalization and Symbolization in Promoting Happiness

According to the social cognitive perspective of moral identity, moral identity usually contains two dimensions: internalization and symbolization [12,13]. The two-dimensional concept of moral identity corresponds to many concepts in social psychology, such as the “having” side and “doing” side of the moral self [38], and the view that people are simultaneously both agent and actor [39].

In predicting moral behavior and prosocial behavior of the individual, internalization, in comparison to symbolization, is generally believed to have more robust effects [40]. However, many studies on the consequences of moral identity focus only on the internalization dimension of moral identity, especially the existing research on the relationship between moral identity and well-being [18,19]. Although symbolization is considered more of self-presentation driven by impression management rather than a valid representation of “having” a moral identity, some studies have pointed to a unique role for symbolization [40,41,42]. When prosocial behavior is expected to be recognized, symbolization will show a stronger positive relation with such behavior for people who report low (compared with high) internalization [42]. Symbolization also plays an important role in charitable giving, whereas internalization does not [40]. Symbolization is a better predictor of public and private prosocial behavior than internalization [41]. 

These previous results illustrate that symbolization reflects the degree to which an individual’s moral identity is expressed through action. Engagement in meaningful behavior is an important avenue for people’s well-being [43]. To capture the full picture of the role that moral identity plays in human well-being, research should examine both internalization and symbolization dimensions. A correlational study of moral identity and well-being has found that symbolization is a stronger predictor of positive well-being than internalization [5], but the different mechanisms of internalization, symbolization, and well-being have not been explored. Few existing studies have explored the relationship between the two dimensions of moral identity and SWB, and no study has examined whether the two dimensions of moral identity predict SWB through identity commitment quality. The current study, therefore, examined the role of two dimensions of moral identity—internalization and symbolization on SWB through the mediation effect of identity commitment quality.

### 1.4. Present Study

In summary, the current study was designed to examine the mediating role of identity commitment quality in the relation between moral identity (internalization and symbolization) and SWB among late adolescent students. The expected relationships among the study variables are shown in Figure 1.

Based on the literature reviewed above, five specific hypotheses were proposed, as follows:

**Hypothesis** **1a:**
*The internalization dimension of moral identity would predict SWB.*


**Hypothesis** **1b:**
*The symbolization dimension of moral identity would predict SWB.*


**Hypothesis** **2:**
*The quality of identity commitment would predict SWB.*


**Hypothesis** **3a:**
*The quality of identity commitment would mediate the link between the internalization dimension of moral identity and SWB.*


**Hypothesis** **3b:**
*The quality of identity commitment would mediate the link between the symbolization dimension of moral identity and SWB.*


## 2. Materials and Methods

### 2.1. Participants and Procedure

A total of 463 college students in central China participated in the current research, of which 44 participants were excluded because they did not pass the attention check. Therefore, this yielded a final sample of 419 participants (M_age_ = 19.4, SD = 1.12; age range 16–25 years), that included 326 females (77.8%, M_age_ = 19.5, SD = 1.17) and 93 males (22.2%, M_age_ = 19.3, SD = 0.93). 

The study protocol was approved by the first author’s university. The participants were recruited primarily from a psychology course at a normal university. None of the participants was majoring in psychology, but all of them had taken an introductory psychology course, as this is required as a compulsory course in higher education. This study used an online survey to collect data. Participants scanned the QR code provided by researchers, and they were required to provide informed consent before formally answering the online questionnaire, which included scales such as moral identity, identity commitment quality, positive and negative experience, life satisfaction, and social demographic questions. All students could withdraw their participation during the survey if they wished to do so.

### 2.2. Measures

#### 2.2.1. Moral Identity

Moral identity was measured by the Self-importance of Moral Identity Questionnaire (SMIQ) [12]. This instrument contains two dimensions: internalization and symbolization. The first dimension reflects the degree to which moral traits are reflected in the self-concept (sample item: “Being someone who has these characteristics is an important part of who I am.”), and the second reflects the degree to which these moral traits manifest in the person’s choices and actions (sample item: “The types of things I do in my spare time (e.g., hobbies) clearly identify me as having these characteristics.”). The measure presented participants with a list of nine characteristics that might describe a moral person (i.e., caring, fair, honesty and kind), then participants were required to imagine a moral person who had these characteristics. Afterward, they were asked to indicate their opinions on a ten item, 5-point Likert-type scale (1 = “strongly disagree,” and 5 = “strongly agree”). The SMIQ exhibited good psychometric characteristics in Chinese context [44]. The Cronbach’s alpha coefficients of internalization and symbolization in current research were 0.81 and 0.67, respectively. 

#### 2.2.2. Identity Commitment Quality

The Questionnaire of Eudaimonic Well-Being (QEWB) was used to assess participants’ identity commitment quality [45]. QEWB has been used as the indicator of quality of identity commitment in previous research [36]. Since there was no Chinese version of the QEWB until the time of our study, we used the back translation procedure to form a Chinese version of the QEWB and examined its structural validity. It is worth noting that the original QEWB was a single dimensional structure, but subsequent factor analysis had difficulty in verifying this single dimensional structure without parceling items [46]. Recent studies have proved that there is a general factor for QEWB by using the bifactor method [47,48]. Therefore, in this study, we adopted the bifactor-ESEM method to verify the existence of a higher order general factor. The results revealed an acceptable fit to the bi-factor model with a general factor: χ^2^ = 236.83, *df* = 116, *p* < 0.001, CFI = 0.94, TLI= 0.90, RMSEA (90% CI) = 0.05 (0.05, 0.06), SRMR = 0.06. After checking the correlation between each item and the general factor, it was found that three items (item 7, item 10, and item 17) were not significantly correlated with the general factor. After deleting these three items, it was found that the model fit was improved significantly: χ^2^ = 161.52, *df* = 74, *p* < 0.001, CFI = 0.96, TLI = 0.92, RMSEA (90% CI) = 0.05 (0.04, 0.06), SRMR = 0.03. The mean scores of 18 QEWB items were used to represent the identity commitment quality. The Cronbach’s alpha of the 18 item Chinese version of the QEWB was 0.81. 

#### 2.2.3. SWB

Scale of Positive and Negative Experience (SPANE) [49] and Satisfaction with Life Scale (SWLS) [50] were used to evaluate three components of SWB: positive affect, negative affect, and life satisfaction [51]. SPANE includes the general items of feelings, such as “positive” and “negative”, so it can reflect more general states such as interest, flow and engagement, and physical pleasure [52]. SPANE contains 12 items, with six items tapping into positive experience, and six items tapping into negative experience. Participants responded to these items on a 5-point Likert-scale ranging from 1 (not at all) to 5 (always). The Chinese version of SPANE has indicated good reliability and validity [53]. The five-item SWLS was used to assesses global life satisfaction [50]. The scale has shown good reliability and validity in Chinese culture [54]. A sample item is “I am satisfied with my life”. Participants rated their agreement from 1 for “strongly disagree” to 7 for “strongly agree”. In our sample, the Cronbach’s alpha of positive experience (α = 0.85), negative experience (α = 0.83), and life satisfaction (α = 0.87) was respectively good. According to the suggestions of previous studies [55], SWB was used as a latent variable reflecting three observed indicators: positive experience, negative experience and life satisfaction.

### 2.3. Analysis Strategy

Data were analyzed by SPSS 25.0 and Mplus 8.3. The correlations between all study variables were performed within SPSS. In the main analysis, our hypothetical model was assessed via the structural equation model (SEM) with latent variables using Mplus. The two-step procedure [56] was followed to examine the mediation effect of identity commitment quality between two subdimensions of moral identity and SWB. First, a confirmatory factor analysis (CFA) of the measurement model was conducted to estimate the extent to which each latent variable was represented by its indicators. To control for the inflated measurement errors caused by multiple items, we created two parcels for each of internalization and symbolization dimensions and three parcels for the quality of identity commitment. SWB were represented by three components: positive affect, negative affect, and life satisfaction [55]. Second, the structural model was examined under the premise that the measurement model was satisfactory. The maximum likelihood method (ML) was used as the estimation method, and a 95% bias-corrected bootstrap wad was used to test the significance of the mediation effect. 

To evaluate the overall fit of the model to the data, several indices were calculated in the current study: RMSEA (root mean square error of approximation), CFI (comparative fit index), TLI (Tucker-Lewis index), and SRMR (standardized root mean square residual). The Chi squared test was also included to compare the estimated models. For the CFI and TLI, the values over 0.90 indicated acceptable fit, while values over 0.95 indicated good fit. For the values of RMSEA and SRMR, which were close to 0.05, indicated an excellent fit, while between 0.05 and 0.08 indicated an acceptable fit [57].

## 3. Results

### 3.1. Descriptive Statistics and Zero-Order Correlation

Table 1 provides descriptive statistics and zero-order correlations among all of the study variables. As indicated, there were significant positive correlations among all the variables. Specifically, both two sub-dimensions of moral identity (internalization and symbolization), positively related to quality of identity commitment and SWB. The quality of identity commitment was significantly associated with SWB. Table 2 presents the gender descriptive statistics and independent sample *t*-test of gender differences for each study variable, and no significant gender differences were found in any of the variables, so gender was excluded from subsequent analyses. 

### 3.2. Measurement Model

CFA analysis was conducted to test the measurement model. The measurement model consisted of four interrelated latent variables (internalization, symbolization, quality of identity commitment, and SWB) with ten observed indicators. The results of CFA analysis revealed a good level of model adjustment: χ^2^ = 75.67, *df* = 29, *p* < 0.01, CFI = 0.97, TLI = 0.95, SRMR = 0.04, RMSEA (90% CI) = 0.06 (0.05, 0.08).

### 3.3. Structural Model

The assumed mediating effect of identity commitment quality between the two dimensions of moral identity and subjective well-being was verified by the structural equation modeling approach. The goodness-of-fit indices for the following two models were compared: (1) the partial mediation model between moral identity and SWB through quality of identity commitment with direct paths from both dimension of moral identity (internalization and symbolization) to SWB (Model 1); and (2) the full mediation model between both dimensions of moral identity (internalization and symbolization) and SWB through the quality of identity commitment with direct path from both dimensions of moral identity to SWB constrained to zero (Model 2). 

The partial mediation model posited the indirect effect of both components of moral identity (internalization and symbolization) on SWB through quality of identity commitment including the direct effects from both dimension of moral identity to SWB. The partial mediation model provided a good fit to the data: χ^2^ = 73.41, *df* = 29, *p* < 0.01, CFI = 0.97, TLI = 0.95, SRMR = 0.04, RMSEA (90% CI) = 0.06 (0.04, 0.08). However, after a thorough analysis of the estimated parameters, it was found that the direct path from the two moral identity dimensions to SWB did not reach a statistically significant level: from internalization to SWB (*β* = −0.01, *p* = 0.59), and from symbolization to SWB (*β* = 0.25, *p* = 0.10). These results indicated that internalization and symbolization may not be directly associated with SWB.

Next, the full mediation model (Model 2) was tested with SEM. The resulting parameters indicated that the full mediation model also fit the data well: χ^2^ = 76.61, *df* = 31, *p* < 0.01, CFI = 0.97, TLI = 0.96, SRMR = 0.04, RMSEA (95% CI) = 0.06 (0.04, 0.08). According to the regression coefficients, all the proposed pathways reached significant levels (*p* < 0.05). When comparing the Model 2 with the Model 1 (Δχ^2^ = 3.20, Δ*df* = 2, *p* = 0.10), the result showed that the differences between these two models was not significant, and both models fit the data well. Based on the principle of parsimony, Model 2 constituted the first choice for explaining the relation between SWB and moral identity through the mediation of identity commitment quality (Figure 2).

The Bootstrap estimation procedure (5000 bootstrapping samples were used) was applied to examine the indirect effects, which are shown in Table 3. As zero is not included within the 95% confidence intervals for both internalization (95% CI = [0.06, 0.32]) and symbolization (95% CI = [0.01, 0.30]), significant indirect effects were obtained; these results indicated internalization and symbolization, respectively, played a significant role in SWB through the mediation of identity commitment quality. We also examined which dimension of moral identity exhibits a more crucial role via indirect effect comparison analysis, as the confidence interval of the difference between the two indirect effects (95% CI = [−0.46, 0.41]) contains zero, this result showed that there was no significant difference between internalization and symbolization. According to these results, both dimensions of moral identity exerted their indirect effects on SWB through quality of identity commitment. 

## 4. Discussion

This study aimed to examine the relation between the two dimensions of moral identity and SWB. Based on eudaimonic identity theory, the present work also investigated the underlying mechanism between this relationship by testing the mediating role of quality of identity commitment in late adolescent college students. Our results revealed that internalization and symbolization of moral identity were positively and significantly correlated with SWB. The two sub-dimensions of moral identity were both independently linked to SWB through the mediation of identity comment quality. 

Our first hypothesis (H1a, H1b) assumed that both internalization and symbolization of moral identity would predict SWB. Our findings revealed that although internalization and symbolization of moral identity were significantly positively correlated with SWB, this relationship was fully mediated by the quality of identity commitment. These results suggested that the relationship between morality and happiness might not be directly related, and that the promotion of morality to happiness mainly originated from the improvement of eudaimonic functioning. Nevertheless, the positive correlation between moral identity and SWB was in line with the view of identity domain that identity formation paves the way for better health and well-being in adolescence [58]. Identity achievers, or those who have formed their identity, have more healthy and positive psychosocial functioning compared with people who have diffusion identity [14,15]. As identity studies have concentrated on identity status and neglected important identity content (e.g., moral identity) in predicting a human being’s healthy and positive psychosocial functioning [14,15,18], our study, by exploring the role of moral identity-an important personal identity-in predicting SWB in late adolescent or early adulthood, filled this gap. In this study, contrary to our expectations, the direct effects of moral identity and subjective well-being did not reach significant levels. Although many studies have showed that moral identity is closely related to well-being [5,18,19], When SWB was used as an indicator of happiness, morality and happiness might not be so closely linked. The following reasons could help explain this. First, SWB is a general and context-free feeling and evaluation of life experience [59]. The self-importance of moral identity is more closely related to eudaimonic well-being in positive psychology, which is a much broader well-being concept that is different from SWB [59,60]. Second, the mechanism of such association between moral identity and SWB needs further exploration based on distinct theoretical support. In this study, we focused on the mechanism of eudaimonic functioning, which was operationalized as identity commitment quality. However, the findings supported the folk concept that people who are moral tend to be happier [2], and people with higher moral identity self-importance are more likely to have higher levels of SWB. A clear explanation of how morality and happiness are linked is more important for promoting the well-being of adolescents.

The hypothesis (H2) that identity commitment quality would be positively correlated with SWB was confirmed. This result was consistent with previous research on identity commitment motivation [35] and identity commitment quality [36]. As suggested by Waterman and colleagues [36] that identity commitment quality plays an important role in the emerging adult’s psychosocial functioning, whereas low-quality commitment is associated with psychological costs. Our findings extended the research on identity commitment quality and well-being by specifically examining the SWB of moral people. Both the self-importance of moral identity, which can be interpreted as moral identity commitment, and the quality of identity commitment played important roles in predicting SWB. This result is consistent with the view of eudaimonic identity theory that the quality of identity commitment, rather than the presence of commitment, predicts positive psychosocial functioning [36]. Thus, the happiness of a moral person may derive from the quality of their moral commitment.

The main goal of our study was to examine the mediating effect of identity commitment quality on the relation between moral identity and SWB (Hypothesis 3a and 3b). To test our hypotheses, we used the QEWB [45] to assess identity commitment quality. The present results suggested that the relationship between moral identity and SWB was fully mediated by identity commitment quality, suggesting that the importance of identity commitment quality was the main way to link moral identity with positive psychological subjective experience. To the best of our knowledge, this study was the first to examine the role of identity commitment quality between moral identity and SWB. Our findings were consistent with existing research on the importance of identity commitment quality [35,36]. According to eudaimonic identity theory [27,28], personally expressive identity is experienced as reflecting one’s “true self”, an experience that is likely to produce eudaimonic well-being. Our findings supported this theory and demonstrated that the important contribution of a personality identity -moral identity -to SWB could be achieved through the quality of identity commitment, as an expression of the individual’s true self. The quality of moral identity commitment is mainly reflected in the fact that it expresses the participants’ purpose and meaning in life as personal expressiveness. This result was also consistent with the eudaimonic activity model [59], which argues that eudaimonic motives/activities may not directly bring well-being, but rather satisfy people’s basic psychological needs by compelling people to do things well, which would lead them to feel well. Moral identity provides such a eudaimonic motive. In our research, the relation between moral identity and SWB was almost indirect when the identity commitment quality was entered, as the direct effect between two dimensions of moral identity (i.e., internalization, symbolization) and SWB was not significant. This pattern of outcomes was largely congruent with prior findings on the link between moral identity and happiness through prosocial behavior [24] and on the significant association between moral identity and prosocial behavior [22]. That is, doing a good deed may not directly lead to happiness; the happiness effect may come from congruence with an individual’s identity [61]. Thus, as moral identity can provide the motivation for moral behavior and contribute to SWB [62], it can be seen as a eudaimonic motive for doing well. This integration of motivation and identity promotes happiness. In summary, these results suggested that high-quality commitment to moral identity provided eudaimonic motivation that ultimately promoted SWB. 

Finally, we distinguished the respective effects of internalization and symbolization on SWB. Although the definition of moral identity contains two conceptual components, namely, internalization and symbolization [12], corresponding to the “having” and “doing” aspects of people’s moral self [38], the existing literature on moral identity and well-being has focused only on internalization and has not adequately captured symbolization. Our results indicated that both internalization and symbolization play an important role in promoting SWB, and both dimensions have an indirect effect on SWB through the mediation of quality of identity commitment. Although some studies have expounded that symbolization has a greater impact on behavior [40,41,42] and well-being [5] compared with internalization, our study compared the indirect effects of internalization and symbolization and found no difference in the effects between the two. In terms of the mechanism by which moral identity leads to SWB, our research finds that both internalization and symbolization act through identity commitment quality.

In conclusion, based on eudaimonic identity theory, the present study shed light on the importance of moral identity and its effect on SWB. First, our study extended the understanding of positive significant associations between moral identity, identity commitment, and one’s SWB. Second, the present study was the first to examine the underlying mechanism of identity commitment quality in mediating the relationship between moral identity and SWB. Third, this study supplemented the lack of attention paid to symbolization of moral identity in research. Like internalization that has been examined in research records for predicting SWB, symbolization was found to predict SWB through the mediation of identity commitment quality. 

### Limitations and Recommendations for Future Studies

This study has several limits that warrant notice. First, the cross-sectional nature of our study could only indicate correlation, as the data were gathered from one-time self-report tools. As such, causal conclusions could not be drawn. Further studies would benefit from using strict experimental designs (such as manipulating the accessibility of moral identity) or adopting longitudinal data to explore the relationship between moral identity and well-being. Second, findings yielded from the college students’ sample may pose threat on the generalizability of our findings. For example, the sample has a gender imbalance, with significantly more women than men, although studies have found that demographic variables like gender have little impact on well-being [45]. Gender imbalance did not prevent us from drawing our conclusions. Third, regarding the relation between identity formation and SWB, we only focused on the role of self-importance of moral identity, and did not measure specific indicators of identity formation. This is because our participants were college students in late adolescence. Although they are still in the integration process of moral identity and self-identity, they have formed a stable identity according to the traditional perspective of identity development. Further studies would best consider both domain-specific and general identity status. Fourth, this study only examined the role of moral identity and identity commitment quality in promoting adolescents’ SWB. Future research could expand the scope of concern about adolescent well-being, such as using specialized measurement of adolescents’ well-being [63] and examining the role of identity commitment and quality on adolescents’ psychological difficulties. Despite these limitations, the present study provides solid evidence for the importance of moral identity and identity commitment quality in predicting college students’ SWB. 

## 5. Conclusions

In conclusion, our results suggested positive and significant associations among moral identity, identity commitment quality, and SWB. Specifically, both the internalization and symbolization dimensions of moral identity predicted SWB through identity commitment quality, and identity commitment quality fully mediated the pathway relationship between moral identity and SWB.

## Figures and Tables

**Figure 1 ijerph-18-09795-f001:**
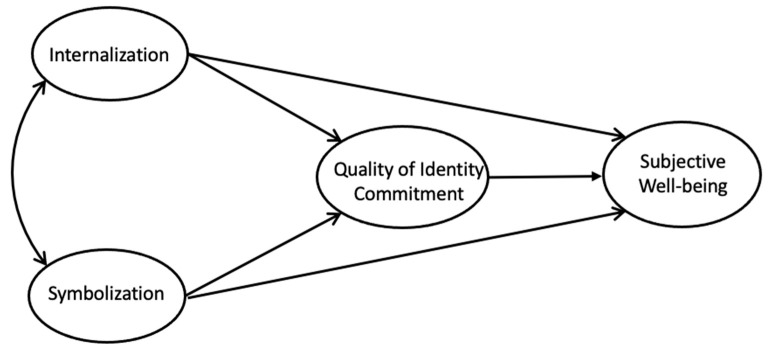
The assumed mediation model.

**Figure 2 ijerph-18-09795-f002:**
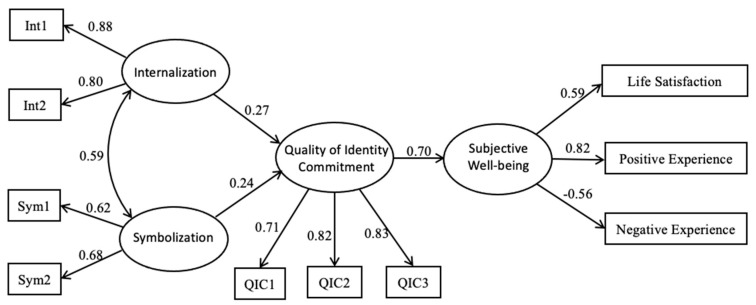
The final mediation model. Note: Factor loadings are standardized. Int1 and Int2 are two parcels of internalization; Sym1 and Sym2 are two parcels of symbolization; QIC1, QIC2 and QIC3 are three parcels of quality of identity commitment; All the path coefficients are significant at the 0.05 level.

**Table 1 ijerph-18-09795-t001:** Descriptive statistics and correlations between study variables.

	M	SD	1	2	3	4	5	6	7
1. Int	4.25	0.52	-						
2. Sym	3.41	0.52	0.41 ***	-					
3. QIC	3.73	0.42	0.34 ***	0.25 ***	-				
4. LS	4.28	1.03	0.13 **	0.19 ***	0.40 ***	-			
5. PE	3.80	0.55	0.24 **	0.26 ***	0.50 ***	0.48 ***	-		
6. NE	2.17	0.57	−0.14 **	−0.05	−0.36 ***	−0.27 ***	−0.48 ***	-	
7. SWB	−0.01	2.35	0.22 ***	0.21 ***	0.54 ***	0.75 ***	0.84 ***	−0.75 ***	-

Note: Int = Internalization of Moral Identity, Sym = Symbolization of Moral Identity, QIC = Quality of identity commitment. LS = Life satisfaction, PE = Positive Experience, NE = Negative Experience, SWB = Z_LS_ + Z_PE_ − Z_NE_. ** *p* < 0.01, *** *p* < 0.001.

**Table 2 ijerph-18-09795-t002:** Independent sample *t* test for gender difference.

Variables	Males (SD)	Females (SD)	*t*	*p*
1. Int	4.24 (0.55)	4.26 (0.51)	0.31	0.76
2. Sym	3.38 (0.50)	3.42 (0.53)	0.64	0.52
3. QIC	3.72 (0.44)	3.74 (0.46)	0.27	0.78
4. LS	4.26 (1.12)	4.29 (1.01)	0.28	0.78
5. PE	3.80 (0.58)	3.80 (0.54)	−0.08	0.94
6. NE	2.09 (0.55)	2.20 (0.57)	1.56	0.12
7. SWB	0.11 (2.43)	−0.05 (2.33)	−0.58	0.56

Note: Variable names are the same as above.

**Table 3 ijerph-18-09795-t003:** Standardized effects and 95% confidence intervals.

Model Pathways	Estimate	SE	Bias Corrected Bootstrap 95%CI		Est./SE	*p*
			Lower	Upper		
Int→QIC→SWB	0.19	0.07	0.06	0.32	2.81	<0.001
Sym→QIC→SWB	0.16	0.08	0.01	0.32	2.05	<0.05
Int→QIC	0.27	0.10	0.08	0.46	2.82	<0.001
Sym→QIC	0.24	0.11	0.02	0.45	2.12	<0.05
QIC→SWB	0.70	0.05	0.60	0.80	13.95	<0.001

Note: Int = Internalization of Moral Identity, Sym = Symbolization of Moral Identity, QIC = Quality of Identity Commitment, SWB = Subjective Well-being.

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
