# Peer review of "Moral Identity and Subjective Well-Being: The Mediating Role of Identity Commitment Quality"

_ijerph, 2021, doi:10.3390/ijerph18189795_

Round 1

Reviewer 1 Report

The manuscript describes a cross-sectional survey on the relationships between moral identity, quality of identity commitment, and subjective well-being. The authors should be commended for the extensive literature review, clear description of the methodology and data analysis, and appropriate reporting of results. The topic seems innovative and interesting. However, there are some issues that should be addressed before it can be accepted for publication.

Introduction

  1. The literature review is very thorough and covered many relevant studies on the variables being examined. However, most of the concepts were not clearly defined before going straight into the discussion about their relations with other variables or mechanisms. For example, moral identity is the main topic in the manuscript, but it was not described or well-defined early on in the introduction (pages 1-3, lines 27-106). The components of moral identity were suddenly introduced without any background information (page 2, lines 66-68). These concepts should be clearly defined before going into deep discussion about them. It was not until after pages of the manuscript that these concepts were introduced in detail (page 4, lines 160-196).
  2. Several theories were briefly introduced. However, the authors do not provide sufficient explanations about them and their relevance for the study. For example, on pages 2-3, lines 95-99, eudaimonic identity theory was simply described as “identity formation can bring eudaimonic well-being rather than hedonic pleasure” and the authors immediately jumped into their hypotheses based on this brief description. Further elaboration is needed. Please see Waterman & Schwartz (2013) Eudaimonic Identity Theory. In The Best Within Us: Positive Psychology Perspectives on Eudaimonia. They provided a detailed discussion on this theory.
  3. For the mediating role of identity commitment quality, more explanation is needed on the differences and overlap between the conceptualizations of eudaimonic well-being, eudaimonic functioning, and identity commitment quality. This clarification is important because eudaimonic well-being is operationalized as identity commitment quality in this manuscript. Waterman & Schwartz (2013) provided a clear discussion of these concepts.
  4. Hypotheses 1a and 1b predicts the relationship between moral identity and SWB. However, these direct paths were not included in Figure 1.

Method

  1. 463 college students were recruited. How was the sample size determined? Was power analysis conducted a priori?
  2. For the Questionnaire of Eudaimonic Well-being (QEWB), the authors stated that “the three-factor solution proposed by Schutte et al. (2013)” was adopted. Please clarify whether this 3-factor model was tested and supported or if the 3 factors were subsumed under a higher-order construct. If it is the latter, why was it tested this way? A higher-order model is different from a 3-factor model.
  3. Similarly, majority of the research conducted on subjective well-being distinguishes the 3 components: positive affect, negative affect, and life satisfaction. Why were they analysed as part of a higher-order construct as opposed to separate components?
  4. As mentioned by the authors in the limitations section, since the sample has a gender imbalance, why were there no controls included for the demographic variables? It was even mentioned that “Future studies can… control for demographic variables”. Why is not done in this study now?
  5. Please specify the number of bootstrap samples used for the indirect effects.

Results

  1. Please summarize the factor loadings of the CFA in text or provide them in a table.

Discussion

  1. The discussion states that “Our results revealed that moral identity was significantly associated with the quality of identity commitment and SWB” and “Our first two hypotheses predicted the positive association of both internalization and symbolization of moral identity with SWB, and our findings supported these hypotheses. Both dimensions of moral identity positively predicted participants’ SWB” (page 9, lines 365-371). This is incorrect. The results clearly showed that the first 2 hypotheses were rejected (page 8, lines 325-329): “it was found the direct path from the two moral identity dimensions to SWB did not reach statistically significant level: from internalization to SWB (B = -.01, p = .59), and from symbolization to SWB (B = .25, p = .10).” This inconsistency needs to be corrected and the non-significant findings should be discussed.

Others

  1. There are some minor grammatical or spelling mistakes in various places. Please do some copyediting.

Reviewer 2 Report

The authors present an interesting, well-founded and well-developed work. They present a well-updated bibliography, almost 40% of the references are from the last 5 years. Methodologically the work is well planned and well resolved and the disclosure is well focused, closely related to what was raised in the introduction and very well adjusted to the data. However, there are a number of issues that I would like to clarify.

1.- There are a series of references that in the text are badly coded, for example, in ln 60, it includes "Garcia et al. (2018)", without coding the corresponding number, it should be "Garcia et al. [ 4] ", the same happens with" Hardy et al. (2013) "or with" Aquino and Reed (2002) ".

2.- The hypotheses are incomplete, a hypothesis 3a appears, and it suggests that at least there should be a 3b, which is not there. But in the discussion a hypothesis 3b is mentioned, which logically is "The quality of identity commitment would mediate the link between the symbolization dimension of moral identity and SWB."

3.- The sample has many more women than men, 326 compared to 93. This data must be duly justified
in the participants section. And in the results section, it should be checked if there are differences in the variables analyzed between men and women, not in the limitations section.

4.- The sample should be better explained, it is said that they are university students, and in the next paragraph that they were recruited via psychology class. In addition, the average age of the group and of the women is provided, the average age of the men should also be reported and it is always appreciated that the age range is reported. It would also be convenient to report other types of variables if information is available. It would also be convenient to make the data collection procedure more explicit, to which page the QR led, what type of information is collected, how does the participant give their informed consent, what type of information is provided about the procedure, etc.

5.- I believe that the results and the discussion should analyze the data in a more exhaustive way and present the mediation effects obtained in a more explicit and clear way.

Reviewer 3 Report

The work entitled “Moral Identity and Subjective Well-being: The Mediating Role of Identity Commitment Quality contains new scientific knowledge and covers a relevant topic. However, I have some comments that have to be addressed before it can be considered for publication.

In the introduction, I think that authors have made a good work in contributing information about previous literature. However, I feel that more recent research about wellbeing during late adolescence in general and specifically the relation between subjective wellbeing and other possible psychological difficulties is missing.

Related to the above mentioned, the discussion section could benefit from more recent literature. In addition, I would add, as limitation, that the sample was composed of psychology students, what precludes the generalization of the results.

Also, I feel that the objectives and hypotheses do not match the results and discussion section. There are not objectives, for instance, for the study of structural equation model.

In the participants’ description, it would be advisable to describe the age distribution. Also, authors should mention if the asked for previous or current history of mental disorder. Also, most of the participants were women, what precludes the generalization of the result to the population.

In 2.2.3. authors should previously explain the meaning of PE and NE: “In our sample, PE α = .85, NE α = .83, SWLS α = .87.”

2.2. Analysis strategy should be 2.3.

Round 2

Reviewer 1 Report

In the results section, lines 325-328, the newly added text: "Table 2 pre-325 sents the gender descriptive statistics and independent sample t-test of gender differences 326 for each study variable, and no significant gender differences was found in any of the 327 variables, so age was excluded from subsequent analyses." There is a typo? it should be "...gender was excluded from subsequent analyses"?

A related question is since the authors stated that the development of moral identity is greatly promoted in late adolescence and early adulthood, doesn't this suggest that age is an important factor in moral development? The age range of the sample is between 16 to 25 years, which covers both adolescence and adulthood. Why was age not accounted for in the analyses?

Reviewer 3 Report

Authors have addressed all my previous suggestions. I have no further comments

Author Response

Thank you very much for your comments!